# Effects of cerebral near-infrared spectroscopy on the outcome of patients undergoing cardiac surgery: a systematic review of randomised trials

Giuseppe Filiberto Serraino, Gavin J Murphy

Department of Cardiovascular Sciences and NIHR Leicester Biomedical Research Unit in Cardiovascular Medicine, University of Leicester, Clinical Sciences Wing, Glenfield General Hospital, Leicester, UK

**Correspondence to**
Dr Gavin J Murphy;
gjm19@le.ac.uk

## ABSTRACT

**Objectives** Goal-directed optimisation of cerebral oxygenation using near-infrared spectroscopy (NIRS) during cardiopulmonary bypass is widely used. We tested the hypotheses that the use of NIRS cerebral oximetry results in reductions in cerebral injury (neurocognitive function, serum biomarkers), injury to other organs including the heart and brain, transfusion rates, mortality and resource use.

**Design** Systematic review and meta-analysis.

**Setting** Tertiary cardiac surgery centres in North America, Europe and Asia.

**Participants** A search of Cochrane Central Register of Controlled Trials, ClinicalTrials.gov, Medline, Embase, and the Cumulative Index to Nursing and Allied Health Literature Plus from inception to November 2016 identified 10 randomised trials, enrolling a total of 1466 patients, all in adult cardiac surgery.

**Interventions** NIRS-based algorithms designed to optimise cerebral oxygenation versus standard care (non-NIRS-based) protocols in cardiac surgery patients during cardiopulmonary bypass.

**Outcome measures** Mortality, organ injury affecting the brain, heart and kidneys, red cell transfusion and resource use.

**Results** Two of the 10 trials identified in the literature search were considered at low risk of bias. Random-effects meta-analysis demonstrated similar mortality (risk ratio (RR) 0.76, 95% CI 0.30 to 1.96), major morbidity including stroke (RR 1. 08, 95% CI 0.40 to 2.91), red cell transfusion and resource use in NIRS-treated patients and controls, with little or no heterogeneity. Grades of Recommendation, Assessment, Development and Evaluation of the quality of the evidence was low or very low for all of the outcomes assessed.

**Conclusions** The results of this systematic review did not support the hypotheses that cerebral NIRS-based algorithms have clinical benefits in cardiac surgery.

**Trial registration number** PROSPERO CRD42015027696.

## INTRODUCTION

Brain injury is a common and severe complication of cardiac surgery with cardiopulmonary bypass (CPB), affecting up to 40% of patients, where it contributes to morbidity, mortality and the increased use of hospital resources.[1]

---

### Strengths and limitations of this study

► We performed a systematic review of randomised trials evaluating the clinical benefits of near-infrared spectroscopy-based cerebral oximetry monitoring during cardiopulmonary bypass.
► This is the most comprehensive review of this technology to date.
► All of the existing trials had important methodological limitations, including importantly a lack of blinding of clinical personnel.
► Grades of Recommendation, Assessment, Development and Evaluation of the quality of the evidence was low or very low for all of the outcomes assessed, indicating a high likelihood that the findings of the review may be altered by subsequent trials.

---

The pathophysiology of CPB-associated brain injury is multifactorial[2] but is thought to involve regional hypoperfusion and tissue hypoxia,[3 4] often within vascular beds, which are already abnormal due to advanced age or comorbidities such as diabetes.[5 6] Previous studies have suggested that cerebral oxygenation may be measured non-invasively using near-infrared spectroscopy (NIRS) sensors applied to the forehead,[7] and more importantly that targeted interventions during bypass that increase NIRS-measured cerebral oxygenation have clinical benefits including reductions in brain injury[8] and reductions in injury to other organ systems as a result of improved overall perfusion.[9 10] It has also been suggested that the use of NIRS may allow the safe application of restrictive red cell transfusion thresholds where there is evidence of adequate cerebral oxygenation, a putative personalised transfusion indicator.[7 11] However, there is uncertainty as to the clinical benefits of NIRS,[12] and this leads to variability in the use of these devices.[13] To address this uncertainty we performed a systematic review and meta-analysis of randomised trials

that had evaluated the effects of personalised NIRS-based algorithms on clinical outcomes, including mortality, organ (brain, heart, kidney) injury, transfusion and resource use.

## METHODS

A systematic review was performed using the methods described in the Cochrane Handbook for Systematic Reviews of Interventions.[14] The review protocol was registered prospectively at http://www.crd.york.ac.uk/PROSPERO/display_record.asp?ID=CRD42015027696, and is included in online supplementary file 1.

### Types of studies

The studies are randomised controlled trials (RCTs) irrespective of blinding, language, publication status, date of publication and sample size.

### Types of participants

Patients undergoing cardiac surgery for acquired or congenital disease or aortovascular disease with or without CPB were included. No age restriction was applied. There were no exclusion criteria.

### Types of interventions

The intervention is a goal-directed NIRS algorithm during cardiac surgery with CPB. The comparator/control is an untreated group or an alternative (non-NIRS-based) goal-directed therapy.

### Types of outcome measure
#### Primary outcomes
Mortality: 30-day or hospital all-cause mortality.

#### Secondary outcomes
1. Acute brain injury: stroke, transient ischaemic attack as defined by study authors.
2. Low cardiac output as defined by study authors.
3. Myocardial infarction as defined by study authors.
4. Acute kidney injury stage 3 or requiring haemofiltration as defined by study authors.
5. Neurocognitive function: group means as described by neurocognitive tests. Tests recommended by a consensus statement to test all key domains of cognitive function[15] are marked with *. Studies were categorised as yes/no as to whether they had assessed the key domains described in the consensus statement. Key domains that were assessed[16] and examples of likely tests that were recorded are as follows:
   – Attention: sustained and divided attention: consensus statements recommend the Trail-Making Test parts A* and B*.[17 18]
   – Verbal memory: consensus statements recommend the Rey Auditory Verbal Learning Test and Rey Auditory Verbal Learning Test*.[18]
   – Visual-spatial: as the Block Design from the Wechsler Adult Intelligence Scale.[19]
   – Psychomotor speed: consensus statements recommend tests such the Digit Symbol Test from the Wechsler Adult Intelligence Scale.[19]
   – Executive function/verbal fluency: consensus statements recommend tests such the Controlled Oral Word Association Test.[20]
   – Motor coordination: consensus statements recommend tests such the Grooved Pegboard Test*, dominant and non-dominant hand.[17]
   – Frontal lobe: saccadic and antisaccadic eye movements.
6. Assessment of important covariates (Mini-Mental State Examination, anxiety and depression, intellectual ability, and concomitant medication) in these analyses were documented.
7. Neurocognitive dysfunction, as reported by the study authors: a consensus definition is a change in a single test of >1 SD; this was defined as either change >1 SD in a group mean (adjusted for baseline) or a change in neurocognitive scores of >1 SD for individual patients;[21] studies were further categorised as defining cognitive dysfunction using a consensus versus a non-consensus definitions.
8. Risk of receiving blood transfusion as defined by study authors.
9. Reoperation for bleeding as defined by study authors.
10. Resource use: intensive care unit (ICU) and hospital length of stay (LOS) as defined by study authors.
11. S100B levels, a marker of brain injury, as reported by study authors.

### Search methods for identification of studies
Potentially eligible trials were identified by searching the Cochrane Central Register of Controlled Trials, Medline, Embase, and the Cumulative Index to Nursing and Allied Health Literature (CINAHL) Plus, using a combination of subject headings and text words to identify relevant trials. The search was performed from inception until November 2016. The Medline search strategy below was adapted as appropriate for other databases:

((Cardiopulmonary Bypass) OR (Cardiac Surgery) OR (Coronary Artery Bypass) OR (Extra Corporeal Circulation) OR (Perioperative Morbidity)) AND ((Near Infra-Red Spectroscopy) OR (Oximetry) OR (Brain/Metabolism) OR (Cerebral Desaturation) OR (Cerebral Perfusion) OR (Cerebral Ischemia) OR (Cerebral Oximetry) OR (Cerebral Saturation) OR (Near Infrared Oximetry) OR (Cognitive)).

To identify ongoing or unpublished trials, we also searched the ClinicalTrials.gov using the following search terms:

Search terms: Randomized
Study type: Interventional studies
Conditions: ((Cardiopulmonary Bypass) OR (Cardiac Surgery) OR (Coronary Artery Bypass) OR (Extra Corporeal Circulation) OR (Perioperative Morbidity))
Interventions: ((Near Infra-Red Spectroscopy) OR (Oximetry) OR (Brain/Metabolism) OR (Cerebral

Desaturation) OR (Cerebral Perfusion) OR (Cerebral Ischemia) OR (Cerebral Oximetry) OR (Cerebral Saturation) OR (Near Infrared Oximetry) OR (Cognitive))

We will also examined the reference lists of eligible trials and reviews. Searches were not restricted by language or publication status.

## Data collection and analysis
The review was performed in accordance with instructions given in the Cochrane Handbook for Systematic Reviews of Interventions.[22]

## Selection of studies
Two reviewers GJM and GFS identified trials for inclusion independently of each other. Excluded studies and the reason for exclusion were recorded.

## Data extraction (selection and coding)
The two authors independently screened the search output to identify records of potentially eligible trials examining the outcomes, the full texts of which were retrieved and assessed for inclusion.

A standardised form was used to extract data from the included studies for assessment of study quality and evidence synthesis. Extracted information included the following:
► Year and language of publication
► Country of participant recruitment
► Year of conduct of the trial
► Study setting: university teaching hospital, non-university teaching hospital
► Study population: inclusion and exclusion criteria
► Sample size
► Participant demographics
► Baseline characteristics
► Type of surgery
► Details of NIRS algorithm (Murkin, non-Murkin[23]) and cointerventions (restrictive vs non-restrictive transfusion thresholds)
► Details of comparator: non-NIRS goal-directed therapy, standard care (protocolised care)
► Outcomes and times of measurement.

The two review authors extracted data independently; discrepancies were identified and resolved through discussion. Missing data were requested from study authors. If there is doubt as to whether trials shared participants completely or partially (with common authors and centres), we contacted the study authors to ascertain whether the study report has been duplicated.

## Risk of bias
The following bias risk domains were assessed as low, uncertain or high based on the instructions given in the in the Cochrane Handbook for Systematic Reviews of Interventions[24]:
► Sequence generation
► Allocation concealment
► Blinding of participant, personnel

► Blinding of outcome assessors
► Incomplete outcome data
► Selective outcome reporting
► Source of funding bias.

Trials were classified as having a low risk of bias if they are graded as being at low risk of bias in all of these domains. The two review authors independently assessed the risk of bias in all of the studies. Discrepancies were resolved by discussion.

## Assessment of reporting bias
Where 10 or more studies were identified for each outcome, we assessed publication bias by the visual assessment of funnel plots and Egger's test.[25]

## Measures of treatment effect
For dichotomous variables, we calculated the risk ratio (RR) with 95% CI. For continuous variables, we calculated the mean difference (MD) with 95% CI for outcomes such as hospital stay, and standardised mean difference (SMD) with 95% CI for quality of life (when different scales were used).

## Dealing with missing data
We perform an intention-to-treat analysis where possible. For dichotomous data presented only as percentages, we estimated frequencies using reported sample sizes for this outcome. For continuous outcomes, if the mean and the SD were not available from the trial report, we sought this information from the trial authors. If this information was still not available, we calculated the mean and SD from median (IQRs) using the software available in Review Manager V.5.

## Assessment of heterogeneity
We anticipated that major sources of clinical heterogeneity would be associated with different patient groups (adults, children, congenital vs acquired disease), the use of different goal-directed NIRS algorithms, the use of cointerventions such as restrictive transfusion thresholds, and differences in the methodology used to assess neurocognitive dysfunction. We explored heterogeneity within each meta-analysis using a $\chi^2$ test with significance set at a p value of 0.10, and expressed the percentage of heterogeneity due to variation rather than to chance as $I^2$.[26] We defined heterogeneity as follows:
► $I^2$, 0%–40%: no or mild heterogeneity
► $I^2$, 40%–80%: moderate heterogeneity
► $I^2$, >80%: severe heterogeneity.

In the presence of severe heterogeneity, meta-analysis was not performed.

## Data synthesis
Meta-analyses were performed using the software package Review Manager V.5.2 and in accordance with the recommendations of the Cochrane Handbook for Systematic Reviews of Interventions.[14] For the primary analysis we compared the results of a random-effects model versus

a fixed-effects model to assess the effects of small studies, and for continuous outcomes, pooled MD or SMD by using the inverse variance method.

### Subgroup analyses

Subgroup analyses were performed in trials in which the Murkin algorithm was used to guide goal-directed therapy versus those that did not, and by participant group: coronary artery bypass grafts versus non-CABG, adults versus children, assessment of neurocognitive function that incorporated tests described in a previous consensus statement and studies that combined the NIRS algorithm with a restrictive red cell transfusion trigger. Test for subgroup differences with Review Manager was used with a p value of <0.05 considered statistically significant.[27]

### Sensitivity analyses

Sensitivity analysis excluded trials with high risk of bias for any of the following: random sequence generation, allocation concealment, blinding of participants, healthcare providers or outcome assessors, incomplete outcome data, attrition and other sources of bias including source of funder.

### Summary of findings

We presented the main results of the review in a 'Summary of findings' table. We included the following outcomes.

► Risk of mortality
► Risk of stroke, myocardial infarction or severe acute kidney injury
► Risk of red cell transfusion
► Neurocognitive impairment
► Resource use: ICU and hospital LOS.

We used GRADEpro software to prepare the 'Summary of findings' table. We judged the overall quality of the evidence for each outcome as 'high', 'moderate', 'low' or 'very low' according to the GRADE (Grades of Recommendation, Assessment, Development and Evaluation) approach.[27] We considered the following:

► Impact of risk of bias of individual trials
► Precision of pooled estimate; inconsistency or heterogeneity (clinical, methodological and statistical)
► Indirectness of evidence
► Impact of selective reporting and publication bias on effect estimate.

### RESULTS
### Description of studies
#### Search results

We identified a total of 17 792 references through electronic searches of the Cochrane Central Register of Controlled Trials (n=1347), Medline (n=9924), Embase (n=6159) and the CINAHL Plus (n=362). From ClinicalTrials.gov, we identified 18 trials that potentially met the inclusion criteria. We have shown the flow of search results in figure 1. We excluded 7909 duplicates, then 9820 clearly irrelevant references, by reading titles and

abstracts. No additional references were identified by reference searching. Of 79 study reports retrieved in full text, we excluded 69 references for the reasons listed in online supplementary figure S1. In total, 11 publications describing nine RCTs fulfilled the inclusion and exclusion criteria provided quantitative data for this review[10 28–35] (online supplementary table S1). We also included data from a recent trial undertaken by our own unit, the results of which are in press.[36]

### Description of excluded studies

Four trials that met our inclusion criteria were excluded after review of the full manuscript (online supplementary table S2). There were two additional reports from one trial: an interim analysis[37] and a post-hoc analysis,[38] which was also reported in full.[10] Another trial compared NIRS values in on-pump versus off-pump CABG patients.[39] A fourth trial by Dullenkopf and colleagues[40] was reported in the abstract as an RCT; however, in this trial NIRS sensors were randomised to placement either on the right or the left forehead and there was no clinical NIRS intervention versus control comparison.

### Included studies

Ten trials that evaluated NIRS-based algorithms in patients undergoing cardiac surgery were identified.[10 28–35]

### Participants

Overall, 1466 participants took part in the 10 trials included in the review. The average age of participants in these trials ranged from 34.6±16.3 to 71±11.2 years, and the proportion of women ranged from 12.5% to 66% in the trials that provided this information.[10 28–35] The proportion of postrandomisation withdrawals ranged from 0% to 4.11%. After withdrawal, 1452 participants were included in the quantitative meta-analyses in this systematic review. Five trials were conducted in patients undergoing CABG only,[10 28 31 32 34] and five trials were conducted in patients undergoing valve or CABG, and valve surgery or other cardiac surgical procedure.[29 30 33 35 36] Further details of participants are listed in online supplementary table S1.

### Intervention

Cerebral NIRS values were measured with the INVOS (Somanetics, Troy, Michigan, USA) in nine trials. In one trial[29] three different devices were used: the INVOS, the FORE-SIGHT (CAS Medical Systems, Branford, USA) and the EQUANOX Classic 7600 (Nonin Medical, Plymouth, USA). In two trials the target NIRS values were >75% of baseline.[10 31] In four trials the targets values were >80%.[29 30 32 34] In one trial the target NIRS values were greater than an absolute NIRS value of 60% or >20% compared with the mean value during pulmonary artery catheter insertion.[35] In two trials the target NIRS value was a combination of >80% of baseline or an absolute measure >50%.[28 33] In one trial the target regional oxygen saturation values were specified

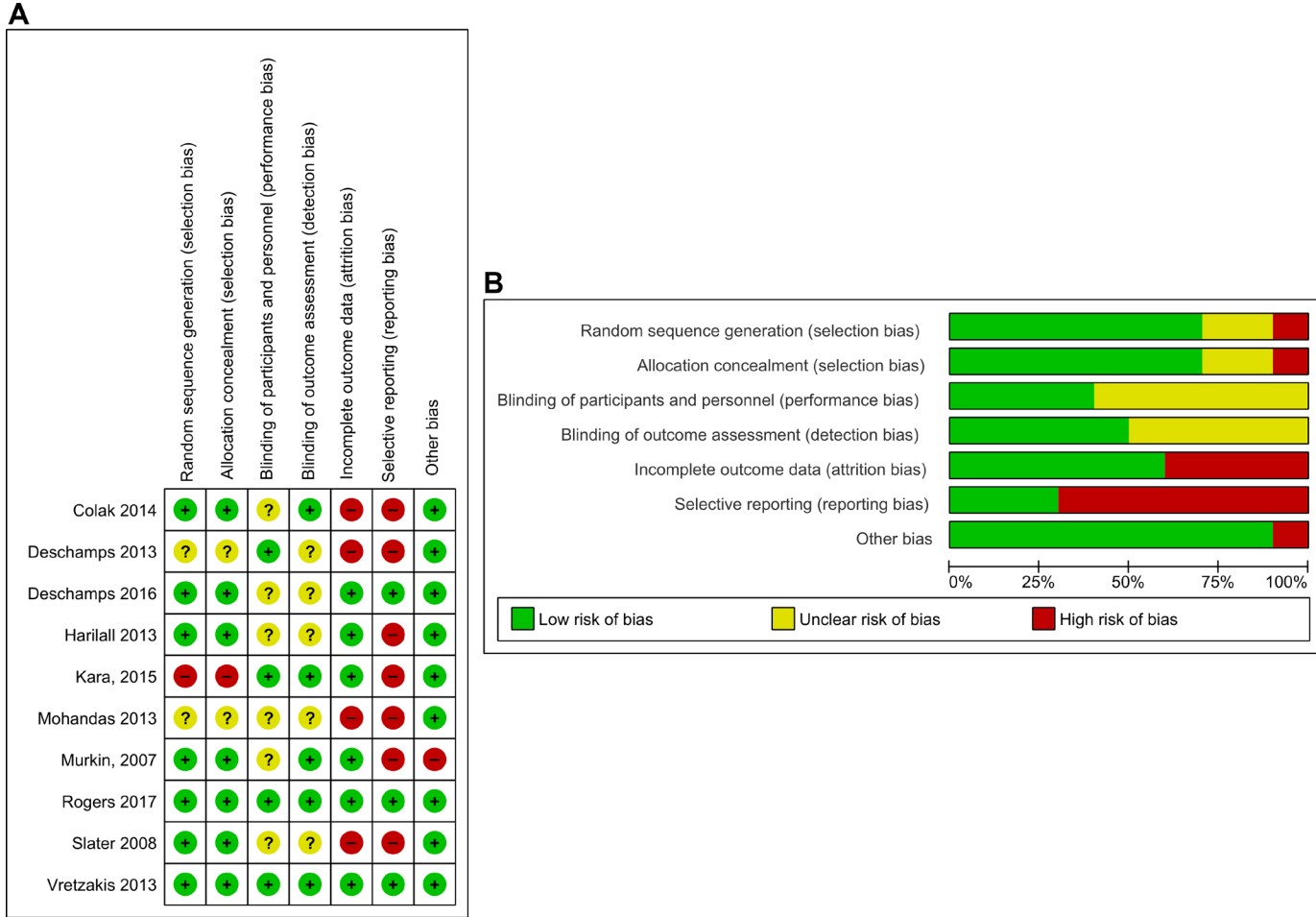

**Figure 1** Risk of bias summaries for (A) individual studies and (B) all studies.

as >70% of preinduction values or an absolute value of >50%.[36] Seven trials used a version of the Murkin algorithm to optimise NIRS values.[10 28–31 33 34 36] Two studies used non-Murkin algorithms.[32 35]

### Control

All trials used standard (protocolised) care as the control group. In seven trials NIRS values were measured in the control group, although the values were hidden from the clinical staff.[10 29–31 33 34 36] No trial considered an alternative patient-specific goal-directed algorithm.

### Cointerventions

In two trials the Murkin algorithm was combined with a restrictive transfusion trigger.[35 36] In these studies a prespecified objective was to determine whether NIRS could be used as part of a patient-specific red cell transfusion indicator.

### Risk of bias in included studies

Risk of bias in individual trials is shown in figure 2A, and the proportions of trials with low risk, unclear risk and high risk of bias in each of the domains are shown in figure 2B. Clinical personnel were not blinded in any trial. However, two trials were considered at low risk of bias in every domain.[35 36]

### Sequence generation

Random sequence generation was adequate in seven trials[10 28 29 31 34–36] and unclear in two trials.[30 33] There was a high risk of bias for random sequence generation in one trial.[32]

### Allocation

Allocation concealment was adequate in seven trials[10 28 29 31 34–36] and unclear in two trials.[30 33] There was a high risk of bias for random sequence generation in one trial.[32]

### Blinding

Theatre staff were unblinded to group allocation in all of the studies. There was evidence of blinding of patients and clinical staff caring for patients postoperatively in four trials[30 32 35 36] and unclear evidence in six trials.[10 28 29 31 33 34] Two trials reported the frequency of protocol compliance.[35 36] In eight trials non-compliance was not monitored or not specified. There was evidence of blinding of outcome assessors in five trials[10 28 32 35 36] and unclear evidence of blinding of outcome assessors in five trials.[29–31 33 34]

### Incomplete outcome data

Six trials reported completeness of follow-up for the primary outcome.[10 29 31 32 35 36] Of these, five trials reported

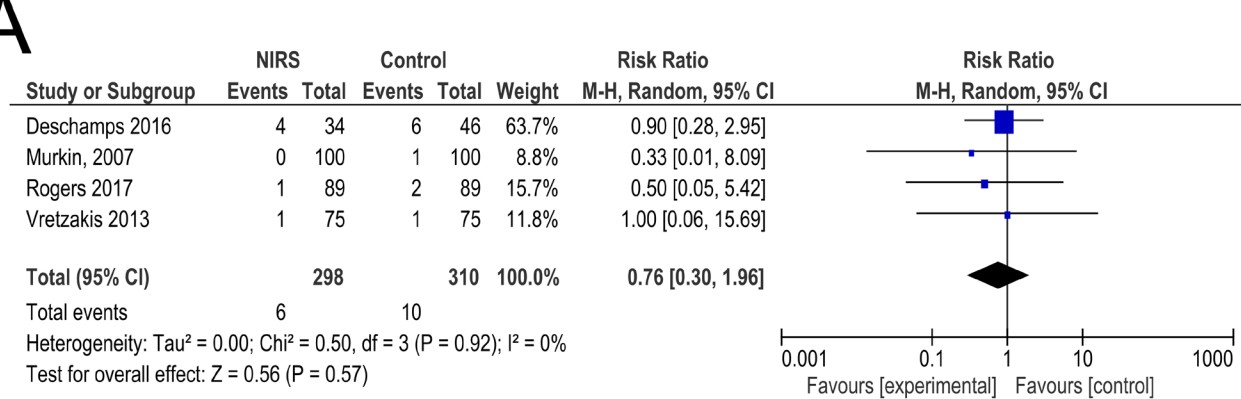

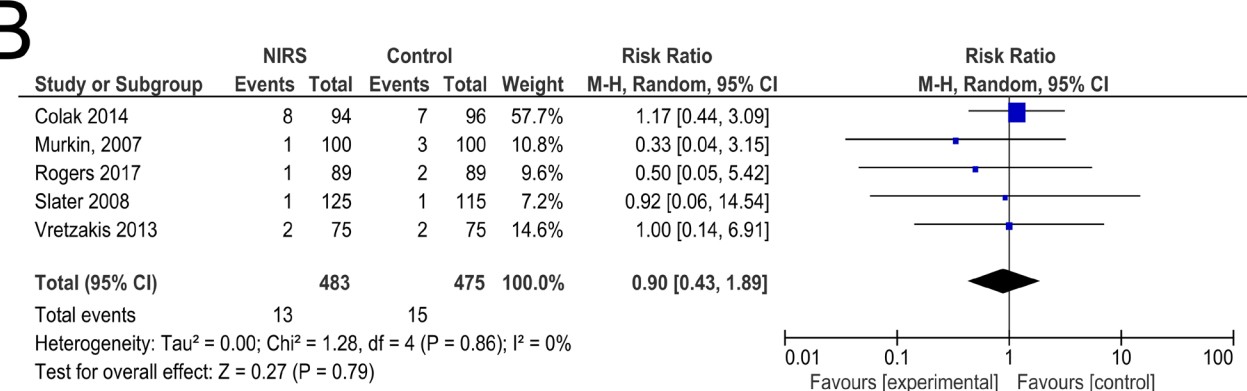

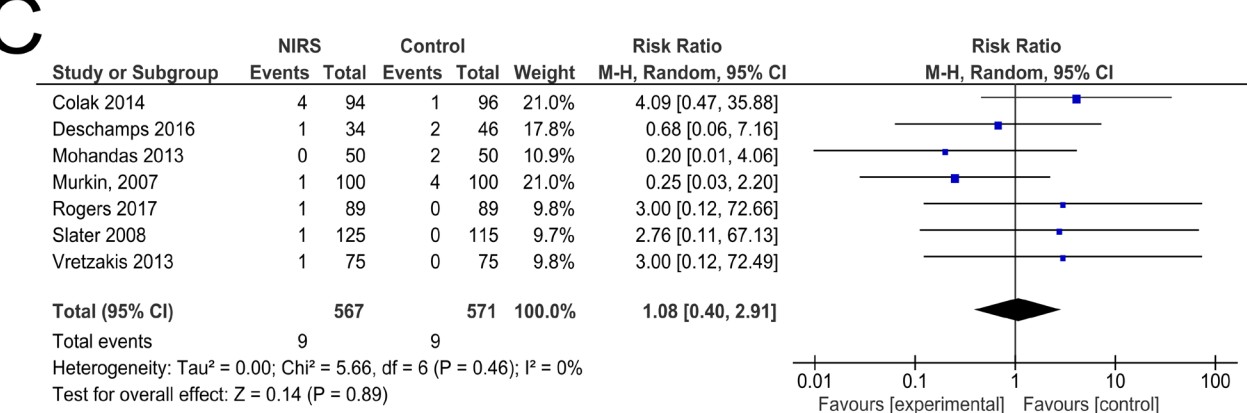

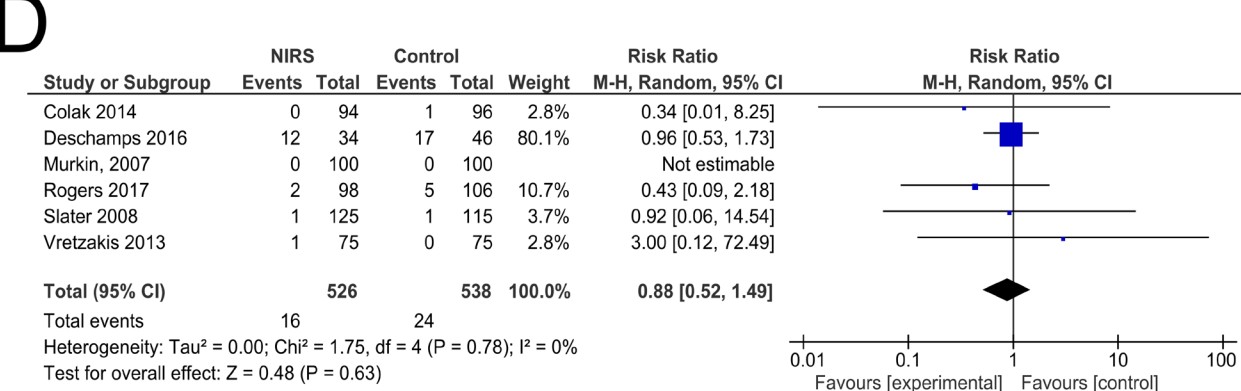

**Figure 2** Forest plots demonstrating summary effect estimates for (A) mortality, (B) myocardial infarction, (C) stroke and (D) stage 3 acute kidney injury or renal replacement therapy. Effect estimates derived using random-effects meta-analysis. Heterogeneity expressed as $\chi^2$ and $I^2$. NIRS, near-infrared spectroscopy.

<10% loss to follow-up, and one trial reported >10% loss to follow-up.[35] Four trials that failed to report completeness of follow-up were considered to be at high risk of attrition bias.[28 30 33 34]

### Selective reporting

Only 1 of the 10 trials included in this review had a published trial protocol.[36] Another five trials had reported details of primary and secondary outcomes in trial registries.[28–30 34 35] Of these, three trials failed to report all the prespecified outcomes.[28 30 34] Of the remaining four trials, none were registered or had published protocols, and were considered at high risk of selective outcome reporting bias.

### Source of funding bias

Sources of funding were reported in six trials. Of these two trials were funded by independent sources. In one trial the study was supported in part by the NIRS device manufacturer and was therefore considered at high risk of funding bias.[10] Three studies that failed to report the source of funding were considered at uncertain risk of funding bias (table 1).

### Effects of intervention

The summary effect estimates for primary and secondary outcomes are described in table 1, figure 2 and online supplementary figure S2.

### Mortality

Four trials with 608 participants reported this outcome.[10 30 35 36] There was no statistically significant difference in mortality between NIRS and controls: RR 0.76, 95% CI 0.30 to 1.96; $I^2$=0%; $\chi^2$ test for heterogeneity p value: 0.92.

### Stroke

Seven trials with 1138 participants reported this outcome.[10 28 29 33–36] There was no statistically significant difference in the frequency of stroke between NIRS and controls: RR 1. 08, 95% CI 0.40 to 2.91; $I^2$=0%; $\chi^2$ test for heterogeneity p value: 0.46.

### Myocardial infarction

Six trials with 1038 participants reported this outcome.[10 28 29 34–36] There was no statistically significant difference in the frequency of myocardial infarction between NIRS and controls: RR 0.90, 95% CI 0.43 to 1.89; $I^2$=0%; $\chi^2$ test for heterogeneity p value: 0.86.

### Severe acute kidney injury

Six trials with 1064 participants reported this outcome.[10 28 30 34–36] There was no statistically significant difference in the frequency of severe acute kidney injury between NIRS and controls: RR 0.88, 95% CI 0.58 to 1.34; $I^2$=0%; $\chi^2$ test for heterogeneity p value: 0.89.

### Red cell transfusion

Four trials with 744 participants reported this outcome.[10 28 35 36] There was no statistically significant

**Table 1** Summary of findings for all primary and secondary outcomes

| Study | Mortality | | MI, n (%) | | Stroke, n (%) | | AKI, n (%) | | Bleeding, n (%) | | Transfusion, n (%) | | ICU LOS (hour), mean (SD) | | Hospital LOS (day), mean (SD) | |
|---|---|---|---|---|---|---|---|---|---|---|---|---|---|---|---|---|
| | NIRS | Control | NIRS | Control | NIRS | Control | NIRS | Control | NIRS | Control | NIRS | Control | NIRS | Control | NIRS | Control |
| Rogers et al[36] | 1 (1) | 2 (1.88) | 1 (1) | 2 (1.88) | 1 (1) | 0 | 2 (2) | 5 (4.71) | 6 (6.1) | 4 (3.77) | 37 (37.7) | 44 (45.8) | 3.3 (0.81) | 3.67 (0.79) | 7.5 (1.15) | 8 (1.17) |
| Deschamps et al[29] | 4 (11.7) | 6 (13) | 0 | 0 | 1 (2.9) | 2 (4.3) | 12 (35.2) | 17 (47.2) | – | – | – | – | 2.99 (2.26) | 0.39 (2.05) | 11 (7.2) | 9.9 (5.8) |
| Kara et al[32] | – | – | – | – | – | – | – | – | – | – | – | – | 1.74 (0.81) | 2.12 (1.05) | 7.15 (1.39) | 7.67 (1.14) |
| Colak et al[28] | – | – | 8 (8.5) | 7 (7.29) | 4 (4.25) | 1 (1) | 0 | 1 (1) | 0 | 1 (1) | 77 (81) | 73 (76) | 2.7 (6.2) | 1.9 (0.9) | – | – |
| Deschamps et al[30] | – | – | 2 (2.66) | 2 (2.66) | – | – | – | – | – | – | – | – | 2.99 (2.26) | 0.39 (2.05) | 7.6 (5.4) | 7.9 (3.2) |
| Vretzakis et al[35] | 1 (1.33) | 1 (1.33) | – | – | 1 (1.33) | 0 | 1 (1.33) | 0 | 1 (1.33) | 1 (1.33) | 51 (68) | 63 (84) | 2.7 (3.8) | 2.7 (3.6) | 10.9 (3.6) | 10.2 (10.7) |
| Mohandas et al[33] | – | – | – | – | 0 | 2 (4) | – | – | – | – | – | – | 1.49 (0.38) | 1.7 (0.49) | – | – |
| Harilall et al[31] | – | – | – | – | – | – | – | – | – | – | – | – | – | – | – | – |
| Slater et al[34] | – | – | 1 (0.8) | 1 (0.86) | 1 (0.8) | 0 | 1 (0.8) | 1 (0.86) | – | – | – | – | – | – | – | – |
| Murkin et al[10] | 0 | 1 (1%) | 1 (1%) | 3 (3%) | 1 (1%) | 4 (4%) | 0 | 0 | 1 (1%) | 1 (1%) | 8 (8) | 10 (10) | 1.25 (0.84) | 1.87 (2.67) | 6.1 (4.4) | 6.9 (5.5) |
| Summary effect estimate RR (95% CI) | 0.76 (0.30 to 1.96) | | 0.90 (0.43 to 1.89) | | 1.08 (0.40 to 2.91) | | 0.88 (0.52 to 1.49) | | 1.23 (0.46 to 3.32) | | 0.93 (0.77 to 1.12) | | 0.00 (−0.44 to 0.44) | | −0.45 (−0.90 to 0.01) | |

AKI, acute kidney injury; ICU, intensive care unit; LOS, length of stay; MI, myocardial infarction; NIRS, near-infrared spectroscopy; RR, risk ratio.

difference in the frequency of red cell transfusion between NIRS and controls: RR 0.93, 95% CI 0.77 to 1.12; $I^2$=51%; $\chi^2$ test for heterogeneity p value: 0.11.

### Reoperation for bleeding

Four trials with 744 participants reported this outcome.[10 28 35 36] There was no difference in the frequency of reoperation for bleeding between NIRS and controls: RR 1.11, 95% CI 0.41 to 3.04; $I^2$=0%; $\chi^2$ test for heterogeneity p value: 0.69.

### Neurocognitive testing and measuring neurocognitive dysfunction

Five trials that recruited 813 patients reported this outcome.[28 32–34 36] One trial[29] reported only the incidence of delirium. Details of these trials are listed in table 2. The Consensus Statement for the Assessment of Neurocognitive Function in Cardiac Surgery[15] recommends that that the following core tests be performed at baseline and up to 3 months postsurgery: the Rey Auditory Verbal Learning Test, the Trail-Making Test Part A and Trail-Making Test Part B, and the Grooved Pegboard Test to assess the neurocognitive domains attention, verbal memory and motor coordination. The consensus statement defines cognitive decline as a difference for the individual of >1 SD from baseline, or a difference of >1 SD between group means, with adjustment for baseline for at least one test. There was significant heterogeneity for this outcome. Only two trials[34 36] measured cognitive function as recommended by the consensus statement. Both trials reported no difference between the groups for neurocognitive function. In one trial no test data were presented.[34] In the other[36] there was a significant difference between the groups for the Controlled Oral Word Association Test, which assesses the domain executive function/verbal fluency; however, this is not a specified core domain in the consensus statement. The three remaining trials used non-consensus testing protocols and non-consensus definitions of neurocognitive decline, and only one tested patients at 3 months postsurgery. Because of the degree of heterogeneity, we did not perform meta-analyses of these outcomes.

### Intensive care unit length of stay

Eight trials with 1051 participants reported this outcome.[10 28–30 32 33 35 36] There were no statistically significant differences in the duration of ICU stay between NIRS and controls, with moderate heterogeneity: RR MD 0.00, 95% CI −0.44 to 0.44; $I^2$=73%; $\chi^2$ test for heterogeneity p value: 0.0005.

### Hospital LOS

Six trials with 761 participants reported this outcome.[10 29 30 32 35 36] Hospital LOS was less in the NIRS group; however, this was not statistically significant: RR MD −0.45, 95% CI −0.90 to 0.01; $I^2$=0%; $\chi^2$ test for heterogeneity p value: 0.83.

### S100B

Two trials reported this outcome in 138 patients.[31 36] One trial with 40 participants measured S100B preoperatively and postoperatively and reported a significant reduction in S100B in NIRS patients: MD −99.87, 95% CI −105.18 to −94.56. The time of the postsurgery sample was not reported. Another with 98 participants measured S100B preoperatively and at four postsurgery time points, on return to ICU and at 6, 12–24, 24–48 and 96 hours. There was no difference between the groups: ratio of geometric means 1.06, 95% CI 0.95 to 1.19, p=0.29. No meta-analysis was performed due to the heterogeneity for this outcome.

### Subgroup analyses

Results of these subgroup analyses for Murkin versus non-Murkin algorithms, CABG versus non-CABG and assessment of neurocognitive function using tests described in a previous consensus statement are shown in online supplementary table S3.

### Publication bias

A funnel plot of SE versus RR for the included outcomes showed an asymmetrical distribution that indicated publication bias. However, since there was an insufficient number of trials providing data (less than 10 studies identified for each outcome), we did not perform this analysis.

### Sensitivity analyses

Fixed-effects models did not materially change the results of our primary analyses. We also conducted sensitivity analyses in two trials identified as being at low risk of bias.[35 36] These trials reported outcomes in 328 participants. Both trials incorporated restrictive red cell transfusion thresholds within the NIRS algorithm. They reported no difference between NIRS and control groups for mortality: RR 0.67, 95% CI 0.11 to 4.08; $I^2$=0%; $\chi^2$ test for heterogeneity p value: 0.71; stroke: RR 3.0 95% CI −0.32 to 28.54; $I^2$=0%; $\chi^2$ test for heterogeneity p value: 1.00; myocardial infarction: RR 0.76, 95% CI 0.17 to 3.41; $I^2$=0%; $\chi^2$ test for heterogeneity p value: 0.66; severe acute kidney injury: RR 0.83, 95% CI 0.44 to 1.54; $I^2$=0%; $\chi^2$ test for heterogeneity p value: 0.42; reoperation for bleeding: RR 1.50, 95% CI 0.48 to 4.62; $I^2$=0%, $\chi^2$ test for heterogeneity p value: 0.75; ICU LOS: RR −0.03, 95% CI −0.69 to 0.62; $I^2$=0%; $\chi^2$ test for heterogeneity p value: 0.95; and hospital LOS: RR −0.20, 95% CI −1.29 to 0.89; $I^2$=0%; $\chi^2$ test for heterogeneity p value: 0.45. Analyses of these trials suggested that the use of NIRS-based algorithms resulted in reductions in red cell transfusion: RR 0.83, 95% CI 0.71 to 0.98; $I^2$=0%; $\chi^2$ test for heterogeneity p value: 0.52. One of these trials, the PASPORT trial,[36] reported the results of neurocognitive assessments and concluded that there was no difference between NIRS and control groups.

### GRADE assessment

A summary of the main findings of the review are presented in table 3. GRADE assessments of the results were either low or very low for all the outcomes, indicating a high likelihood that these conclusions may be altered by subsequent trials.

**Table 2** Assessment of neurocognitive dysfunction

| Domain | Test protocol | Kara et al[32] | Colak et al[28] | Rogers et al[3 6] | Mohandas et al[33] | Slater et al[34] |
|---|---|---|---|---|---|---|
| Timing | | Baseline and and predischarge | Baseline and 7 days | Baseline, 7 days and 3 months | Baseline, 7 days and 3 months | Baseline, predischarge and 3 months |
| Attention | Trail-Making Test parts A and B, or equivalent | | X (CTT) | X | | X (SCW) |
| Verbal memory | Rey Auditory Verbal Learning Test (VLT) or Hopkins VLT | | | X | | X |
| Visuospatial | Block Design from the Wechsler Adult Intelligence Scale | | | X | | |
| Psychomotor speed | Digit Symbol Test from the Wechsler Adult Intelligence Scale | | | X | | |
| Executive function/verbal fluency | Controlled Oral Word Association Test | | | X | | |
| Motor coordination | Grooved Pegboard Test | | X | X | | X |
| Other | Saccadic and antisaccadic eye movements | | X | X | X | X |
| | Montreal Cognitive Assessment | X | | | | |
| Measurement of confounders | MMSE | | X | X | X | X |
| | Anxiety depression | | | X | | X |
| | Adult Reading Test | | | X | | |
| | Medication | | | X | | |
| | Delirium Rating Score | | | | | X |
| Presented test data | | X | | X | X | |

**Table 2** Continued

| Domain | Test protocol | Kara et al[32] | Colak et al[28] | Rogers et al[3 6] | Mohandas et al[33] | Slater et al[34] |
|---|---|---|---|---|---|---|
| | Timing | Baseline and and predischarge | Baseline and 7 days | Baseline, 7 days and 3 months | Baseline, 7 days and 3 months | Baseline, predischarge and 3 months |
| | Definition of cognitive decline | The maximum score to get from this test is 30: >25= normal, 19–25=mild cognitive impairment, <19= serious cognitive impairment | Dichotomous: decrease in an MMSE score for three or more points from baseline value AND decrease of 1 SD or more in performance on CTT 1 and GP tests | Difference in group mean (p<0.05) in >3 of 6 domains | Postoperative MMSE impairment was defined as a decrease in scores by more than 20% of the preoperative values. Postoperative ASEM impairment was defined as a decrease of scores to more than 30% of preoperative values. | Cognitive decline was defined as a decline of 1 SD or more in performance on one or more of the neuropsychological tests. |
| Reported difference in cognitive function | | X | X | | X | |
| | Estimate of difference | Mild cognitive impairment: 7/43 NIRS, 16/36 control, p=0.01; severe cognitive impairment: 0/42 NIRS, 3/36 control, p=0.09 | 28% NIRS vs 52% control, p=0.0002 | Significant difference in COWAT: mean difference 3.73 (95% CI 1.5 to 5.96), p=0.0011 | ASEM at 3 months: mean 15.69 (SD 3.99); NIRS: mean 17.68 (SD 1.79), p<0.001 | 58% NIRS vs 61% control |

ASEM, antisaccadic eye movements; COWAT, Controlled Oral Word Association Test; CTT, Colour Trail Test 1; GP, Grooved Pegboard Test; MMSE, Mini-Mental State Examination; NIRS, near-infrared spectroscopy; SCW, Stroop Colour and Word.

**Table 3** Summary of main findings of systematic review and GRADE assessment of trial results

**Near-infrared spectroscopy algorithm compared with control (standard care) in cardiac surgery**

**Patient population: adult cardiac surgery; setting: tertiary cardiac centres**

**Intervention: near-infrared spectroscopy algorithms for personalised optimisation of cerebral oxygenation**

**Control: standard care**

| Outcomes | Anticipated absolute effects* (95% CI) | | Relative effect (95% CI) | Participants (n) (studies) | Quality of the evidence (GRADE) |
|---|---|---|---|---|---|
| | Risk with control | Risk with NIRS | | | |
| Mortality | 32 per 1000 | 25 per 1000 (10 to 63) | RR 0.76 (0.30 to 1.96) | 608 (4 RCTs) | ⊕⊕◯◯ Low |
| Red cell transfusion | 504 per 1000 | 469 per 1000 (388 to 564) | RR 0.93 (0.77 to 1.12) | 744 (4RCTs) | ⊕⊕◯◯ Low |
| Stroke | 16 per 1000 | 17 per 1000 (6 to 46) | RR 1.08 (0.40 to 2.91) | 1138 (7 RCTs) | ⊕◯◯◯ Very low |
| Myocardial infarction | 29 per 1000 | 26 per 1000 (12 to 54) | RR 0.90 (0.43 to 1.89) | 1038 (6 RCTs) | ⊕◯◯◯ Very low |
| Renal failure | 71 per 1000 | 62 per 1000 (41 to 95) | RR 0.88 (0.58 to 1.34) | 1043 (6 RCTs) | ⊕◯◯◯ Very low |
| Reoperation for bleeding | 19 per 1000 | 21 per 1000 (8 to 56) | RR 1.11 (0.41 to 3.04) | 744 (4 RCTs) | ⊕◯◯◯ Very low |
| ICU length of stay (ICU LOS) | | The mean ICU LOS in the intervention group was 0 (0.44 lower to 0.44 higher). | | | ⊕◯◯◯ Very low |
| Hospital length of stay (H LOS) | | The mean H LOS was 0.45 lower (0.9 lower to 0.01 higher). | | | ⊕⊕◯◯ Low |

*The risk in the intervention group (and its 95% CIs) is based on the assumed risk in the comparison group and the relative effect of the intervention (and its 95% CI).

GRADE, Grades of Recommendation, Assessment, Development and Evaluation; NIRS, near-infrared spectroscopy; RCT, randomised controlled trial; RR, risk ratio.

## DISCUSSION
### Main findings

A systematic review and meta-analysis of existing trials did not demonstrate clinical benefits attributable to the use of personalised NIRS-based algorithms during CPB. The use of NIRS did not result in reductions in mortality, injury to the brain, heart or kidneys, or reductions in resource use. A qualitative review of studies that had evaluated the effects of NIRS on neurocognitive function did not show clear evidence of benefit. An analysis of two trials at low risk of bias where NIRS was applied along with a restrictive red cell transfusion threshold demonstrated a reduction in red cell transfusion with this approach, with no difference between NIRS treated and controls with respect to clinical outcomes or resource use. Overall the GRADE quality of the evidence was low or very low for all of the outcomes measured.

### Strengths and weaknesses

This is the most comprehensive evaluation of NIRS-based, patient-specific, goal-directed algorithms in cardiac surgery to date. The results supersede those of a previous quantitative review of NIRS in cardiac surgery that considered randomised (2 studies) and observational analyses (27 studies).[41] Our searches also identified a Cochrane review protocol with similar aims; however, the results of this review have yet to be published.[42] The current review used comprehensive search strategies in a wide range of registries and data sources, had access to the full texts of all identified trials, used contemporary risk of bias assessments (GRADE), and assessed a wide range of outcomes after cardiac surgery. The main limitation of the review is that we did not have access to all of the source data. Although additional unpublished information was also obtained from three authors and included in the meta-analysis, we realise that consistent analyses of all studies can only be done when data on individual patients are combined. In addition, the review identified important limitations of existing data; all of the 10 RCTs had limitations in terms of methodological quality. The risk of procedural bias was high in these trials as there was no blinding of clinical personnel. Furthermore only 2 from 10 trials attempted to define the likelihood of

procedural bias by describing the degree of protocol adherence. The reporting of outcomes was also heterogeneous between trials, limiting the number of studies that could be included in each outcome. This was most evident for the outcome cognitive function. Many trials did not report important clinical outcomes, such as death, although it is highly likely that this outcome was measured. Furthermore, in many cases, although the trial report stated that important assessments had been made, particularly with respect to the testing of cognitive function, the results of these assessments were not reported. Despite these limitations the results were remarkably consistent, with low or no heterogeneity for all of the analyses, all of which suggested that there are no clinical benefits attributable to the use of NIRS-based algorithms.

### Clinical importance

NIRS technology and NIRS-based algorithms are used in cardiac surgery centres worldwide, although there is clear evidence of equipoise with respect to the clinical benefits of this technology.[12 13] The results presented here do not support the hypotheses that the use of NIRS may reduce brain injury,[8] or by using the brain as the index organ, reducing injury to the heart or kidneys as a result of improved overall perfusion during cardiac surgery with CPB.[9] They do not provide insights into the role of NIRS in procedures that require deep hypothermic circulatory arrest. It is possible that our results are attributable to chance; our GRADE assessment of the systematic review results was very low for all the prespecified outcomes, indicating a high likelihood that these conclusions may be altered by subsequent trials. There was almost no heterogeneity for these outcomes, however, and these findings were consistent with the results of the PASPORT trial.[36] This multicentre trial recruited a larger cohort than almost all the previous trials, was at low risk of bias, and demonstrated no benefit for NIRS-based algorithms for a range of outcomes, including cognitive function, and biomarkers of myocardial, renal and neurological injury.[36]

Our systematic review indicated that the combination of NIRS and a restrictive transfusion threshold resulted in a reduction in red cell transfusion, without any difference between NIRS treated patients and controls with respect to clinical outcomes or resource use. This may be interpreted as showing that NIRS may be used safely to implement restrictive transfusion thresholds. These findings must be interpreted with caution, however; these two trials enrolled only 387 patients, clinical staff were not blinded to group allocation in these trials, and endpoints such as red cell transfusion are susceptible to performance bias. A final consideration is that the review did not identify any trial in paediatric cardiac surgery, a common setting for the use of these devices, and thus has identified a knowledge gap with respect to the utility of this intervention in these patients.

## CONCLUSIONS

Existing evidence suggests that the use of NIRS-based, patient-specific algorithms that aim to optimise cerebral oximetry does not result in reductions in mortality, major morbidity or resource use in adult cardiac surgery. Assessment of the quality of the evidence indicates that there is a need for further randomised trials at low risk of bias to assess the clinical utility of NIRS in both adult and paediatric cardiac surgery. To determine the clinical utility of these devices, future studies should be designed to evaluate pragmatic, clinically important outcomes such as freedom from death and disability, or major morbidity (stroke, renal failure requiring dialysis, low cardiac output).

**Acknowledgements** The authors gratefully acknowledge the assistance of Mr Giovanni Mariscalco, who reviewed the study protocol for important intellectual content.

**Contributors** Both of the study authors had full access to all of the data (including statistical reports and tables) in the study and can take responsibility for the integrity of the data and the accuracy of the data analysis. GJM conceived and designed the study, and undertook the systematic review. GFS wrote the protocol, undertook the systematic review, performed the analysis and drafted the report along with GJM. Both authors reviewed the report for important intellectual content and approved the final version.

**Funding** This work was supported by the National Institute for Health Research Programme Grants for Applied Health number RP-PG-0407-10384. GFS is supported by the Leicester NIHR Cardiovascular Biomedical Research Unit. GJM is supported by British Heart Foundation Grants RG/13/6/29947 and CH/12/1/29419.

**Competing interests** GFS declares no conflict of interest. GJM reports grants from British Heart Foundation during the conduct of the study and grants from British Heart Foundation, grants from National Institute for Health Research, grants from Zimmer Biomet, personal fees from AbbVie, and personal fees from Thrasos, outside the submitted work.

**Provenance and peer review** Not commissioned; externally peer reviewed.

**Data sharing statement** No additional data available.

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
