## [Reviewer comments · BMJ Open]

ARTICLE DETAILS

TITLE (PROVISIONAL)	Effects of cerebral Near-Infrared Spectroscopy on the outcome of patients undergoing cardiac surgery: a systematic review of randomised trials.
AUTHORS	Serraino, Giuseppe; Murphy, Gavin

VERSION 1 - REVIEW

REVIEWER	Seema Agarwal Liverpool Heart and Chest Hospital UK My spouse is currently the principal investigator in a trial of INVOS in cardiac surgery.
REVIEW RETURNED	20-Mar-2017

GENERAL COMMENTS	REVIEW OF PAPER Major comments : This is a well written review of an important topic in cardiac surgery however as it stands this reviewer has some concerns which make it impossible to accept. 1. There is repeated referral to a study conducted in the author's own institution which apparently has low risk of bias, complete outcome data etc. However the data and results from this trial have not been published and are not available to the reviewer. The reference for this trial is apparently a protocol however entering the details provided does not produce a result in Pubmed. A trial protocol for a similar study is available at JMIR ResProtocol 2015 – this may or may not be the trial which is being referred to however this is not referred to in the paper. The publication does also not contain any results. 2. page 8, line 22/23 “theatre staff were unblinded in all of the studies”. This is not true. In Murkin's study in the control group the screen to the INVOS was switched off whilst still recording – electronic blinding. In Slater's study there was again a blinded control group – the values on the monitor were not displayed or available for the clinician to use in theatre. If the authors mean that the theatre staff were aware of which group the patient was assigned to surely this was necessary if the theatre staff were to follow an algorithm to improve cerebral oxygenation in study patients? 3. page 8, line 48 “The remaining trials were considered at high risk of selective outcome reporting bias” – what is the basis for this
--

	statement? Minor comments : P6, line 43 : rephrase please. Do you mean that "There were two additional reports of one study (an interim analysis and a post hoc analysis) of a trial which was also reported in full." P6, line 47 off pump not of pump P7 Intervention section : there are 10 trials in total you are looking at – you mention target NIRS values in 9 only. You also mention the algorithms followed in 9 trials only. P 8, line 57 “ in one trial the study was funded in part by the NIRS device manufacturer and was therefore considered at high risk of funding bias” – what is the basis for this statement?
--	--

REVIEWER	Andrew Cook University of Southampton, UK
REVIEW RETURNED	12-Apr-2017

GENERAL COMMENTS	I found this an interesting paper. While not a clinical specialist in this area, the importance of the question addressed seems to be well justified. The methods used were appropriate and sound, and the reporting through. The approach used appears to match what was described in the registration, though Giovanni Mariscalco has moved from an author of the protocol to being acknowledged in this paper. The funding source is not consistent between the PROPSERO registration (Leicester Cardiovascular Biomedical Research Unit) and the paper (National Institute for Health Research Programme Grants for Applied Research). The lack of publication for one trial included in the review (PASPORT - reference 26) would make reproducibility a challenge. In addition PASPORT is one of the more influential trials in the review, but the results would appear not to have undergone peer review themselves. It's unlikely the numbers are likely to change following peer review but this does represent a risk to the conclusions here, I'd like to see some reassurance that it will be published reasonably promptly. As it's PGfAR I expect it to appear in the NIHR journals library at some point. I'd like to see the need for future primary research spelt out in more detail. At the moment the suggestion is that studies should be powered to assess the effect of NIRS on important clinical outcomes - that's rather obvious. I'd like to be told what those outcomes are, and if possible what effect sizes should be sought. With regards to the PRISMA checklist - Item 6: I'm not convinced the rationale for the various items here is sufficiently described.
---

	- Item 7: While the databases for the search are identified, the earliest date searched is not. IF from inception, this should be stated. Otherwise all the PRISMA items appear to be adequately addressed. I don't understand the need for the 'online only digital supplement'. BMJ Open is online only with, I understand, no word count limits. It seems that the material in the supplement could appear in the main paper - and certainly as a consumer of systematic reviews that's what I'd prefer. Ultimately of course presentation like this is a matter for the editor. Overall this is a well delivered and well described review, with no fundamental flaws.
--	--

REVIEWER	Ibrahim Kara Sakarya University, Faculty of Medicine/Department of Cardiovascular Surgery/Sakarya, TURKEY
REVIEW RETURNED	25-Apr-2017

GENERAL COMMENTS	We are sending our greetings to the authors for this meta-analyze. NIRS based algorithms are widely used in the group A surgical procedures even in cardiac surgery. Authors have written that NIRS based algorithms have got no beneficial affect on reducing stroke and major complications in open cardiac surgery. Authors have to clear some points in the manuscript. 1- Most of the 10 randomized controlled trials that included to the meta-analyze consists of only CABG surgery patients (Aauthors have excluded high risk patients in this trials) Cardiac surgery is not consist of only CABG surgery, If you mention cardiac surgery it has got a wide spectrum of surgical procedures. For example; Surgery of aortic aneurysms (elective and emergency), Aortic dissection surgery, Valve surgery etc. Authors have included mostly CABG surgery including trials in this meta-analyze so in the conclusion word they can mention only CABG surgery patients. Because the other open heart surgery procedures especially Aortic aneursym and dissection surgery; there is a high risk of complications as stroke and major bleeding. NIRS based algorithms beneficial affect have shown in this open heart surgery procedures in literature. So Authors have to make a revision in the conclusion and discussion part. If the meta-analyze will be as this way, Authors have to mention "Most of the randomized controlled trials in the meta-analyze have consist of only CABG patients" in the Limitations.
--

VERSION 1 – AUTHOR RESPONSE

Reviewer: 1

Reviewer Name: Seema Agarwal

Institution and Country: Liverpool Heart and Chest Hospital, UK.

Please state any competing interests or state 'None declared': My spouse is currently the principal

investigator in a trial of INVOS in cardiac surgery.

Please leave your comments for the authors below REVIEW OF PAPER

Major comments:

This is a well written review of an important topic in cardiac surgery however as it stands this reviewer has some concerns which make it impossible to accept.

1. There is repeated referral to a study conducted in the author's own institution which apparently has low risk of bias, complete outcome data etc. However the data and results from this trial have not been published and are not available to the reviewer. The reference for this trial is apparently a protocol however entering the details provided does not produce a result in Pubmed. A trial protocol for a similar study is available at JMIR ResProtocol 2015 – this may or may not be the trial which is being referred to however this is not referred to in the paper. The publication does also not contain any results.

Thank you for this comment. The PASPORT trial report is now in press and the citation has been altered to reflect this.

2. Page 8, line 22/23 "theatre staff were unblinded in all of the studies". This is not true. In Murkin's study in the control group the screen to the INVOS was switched off whilst still recording – electronic blinding. In Slater's study there was again a blinded control group – the values on the monitor were not displayed or available for the clinician to use in theatre. If the authors mean that the theatre staff were aware of which group the patient was assigned to surely this was necessary if the theatre staff were to follow an algorithm to improve cerebral oxygenation in study patients?

We interpreted blinding as per the second sentence; i.e. clinical staff will have known which group the patient was allocated to. This is a common limitation of device trials, and we believe a potential source of bias. We have revised the manuscript to include the term "theatre staff were unblinded to group allocation in all of the studies"

3. Page 8, line 48 "The remaining trials were considered at high risk of selective outcome reporting bias" – what is the basis for this statement?

Thank you for this comment. The statement has now been qualified as "Of the remaining four trials, none were registered or had published protocols, and were considered at high risk of selective outcome reporting bias"

Minor comments:

P6, line 43: rephrase please. Do you mean that "There were two additional reports of one study (an interim analysis and a post hoc analysis) of a trial which was also reported in full."

Thank you for this comment. The text has been revised as suggested.

P6, line 47 off pump not of pump

The typo has been corrected.

P7 Intervention section: there are 10 trials in total you are looking at – you mention target NIRS values in 9 only. You also mention the algorithms followed in 9 trials only.

The sentence 'In 3 trials the targets values were >80%.29,30,32,34' has now been corrected to 'In 4 trials the targets values were >80%.29,30,32,34' to correct this error.

P 8, line 57 " in one trial the study was funded in part by the NIRS device manufacturer and was therefore considered at high risk of funding bias" – what is the basis for this statement?

This was based on the statements in the primary trial report as follows; 'Supported in part by Canadian Institutes of Health Research grant MOP37914, and a grant from Somanetics Corporation' and 'Mr. R.A. Widman, an employee of Somanetics Corporation, assisted with collection of NIRS data as well as statistical analysis of the data'. Somanetics is the device manufacturer. Table S1 has been

revised to reflect this.

Reviewer: 2

Reviewer Name: Andrew Cook

Institution and Country: University of Southampton, UK.

Please state any competing interests or state 'None declared': None declared

Please leave your comments for the authors below

I found this an interesting paper. While not a clinical specialist in this area, the importance of the question addressed seems to be well justified.

The methods used were appropriate and sound, and the reporting through.

Thank you for this comment.

The approach used appears to match what was described in the registration, though Giovanni Mariscalco has moved from an author of the protocol to being acknowledged in this paper. The funding source is not consistent between the PROPSERO registration (Leicester Cardiovascular Biomedical Research Unit) and the paper (National Institute for Health Research Programme Grants for Applied Research).

Thank you, Mr Mariscalco did not meet the criteria for authorship of this report. The funding statement in the submitted paper is correct.

The lack of publication for one trial included in the review (PASPORT - reference 26) would make reproducibility a challenge. In addition PASPORT is one of the more influential trials in the review, but the results would appear not to have undergone peer review themselves. It's unlikely the numbers are likely to change following peer review but this does represent a risk to the conclusions here, I'd like to see some reassurance that it will be published reasonably promptly. As it's PGfAR I expect it to appear in the NIHR journals library at some point.

Thank you for this comment. As per our response to reviewer 1, we have now added a citation referring the imminent publication of the PASPORT trial in the British Journal of Anaesthesia.

I'd like to see the need for future primary research spelt out in more detail. At the moment the suggestion is that studies should be powered to assess the effect of NIRS on important clinical outcomes - that's rather obvious. I'd like to be told what those outcomes are, and if possible what effect sizes should be sought.

Thank you for this comment, this section of the discussion has been expanded as suggested.

With regards to the PRISMA checklist

- Item 6: I'm not convinced the rationale for the various items here is sufficiently described.

The eligibility criteria from the eSupplement have been added to the main paper.

- Item 7: While the databases for the search are identified, the earliest date searched is not. IF from inception, this should be stated.

The text has been revised to include the term 'from inception'

Otherwise all the PRISMA items appear to be adequately addressed.

I don't understand the need for the 'online only digital supplement'. BMJ Open is online only with, I understand, no word count limits. It seems that the material in the supplement could appear in the main paper - and certainly as a consumer of systematic reviews that's what I'd prefer. Ultimately of course presentation like this is a matter for the editor.

Thank you for this comment, as per our response to the Editor these supplementary methods have now been added to the main paper.

Overall this is a well delivered and well described review, with no fundamental flaws.

Reviewer: 3

Reviewer Name: Ibrahim Kara

Institution and Country: Sakarya University, Faculty of Medicine/Department of Cardiovascular Surgery/Sakarya, TURKEY Please state any competing interests or state 'None declared': Non declared

Please leave your comments for the authors below.

We are sending our greetings to the authors for this meta-analyze. NIRS based algorithms are widely used in the group A surgical procedures even in cardiac surgery. Authors have written that NIRS based algorithms have got no beneficial affect on reducing stroke and major complications in open cardiac surgery. Authors have to clear some points in the manuscript.

1- Most of the 10 randomized controlled trials that included to the meta-analyze consists of only CABG surgery patients (Authors have excluded high risk patients in this trials)

Cardiac surgery is not consist of only CABG surgery, If you mention cardiac surgery it has got a wide spectrum of surgical procedures. For example; Surgery of aortic aneurysms (elective and emergency), Aortic dissection surgery, Valve surgery etc. Authors have included mostly CABG surgery including trials in this meta-analyze so in the conclusion word they can mention only CABG surgery patients. Because the other open heart surgery procedures especially Aortic aneursym and dissection surgery; there is a high risk of complications as stroke and major bleeding. NIRS based algorithms beneficial affect have shown in this open heart surgery procedures in literature.

So Authors have to make a revision in the conclusion and discussion part. If the meta-analyze will be as this way, Authors have to mention "Most of the randomized controlled trials in the meta-analyze have consist of only CABG patients" in the Limitations.

Thank you for this comment. Only four of the 10 trials identified in the review were restricted to CABG patients. We therefore do not fully agree with this statement from the reviewer. However, we have now stated in the discussion that the trial does not reflect the use of NIRS devices in trials where deep hypothermic circulatory arrest is used.

Therefore, on behalf of my co-author, I am submitting the revised manuscript for consideration and possible publication. All authors have read and approved the submitted manuscript, the manuscript has not been submitted elsewhere nor published elsewhere in whole or in part.

VERSION 2 – REVIEW

REVIEWER	DR SEEMA AGARWAL LIVERPOOL HEART AND CHEST HOSPITAL
REVIEW RETURNED	06-Jun-2017

GENERAL COMMENTS	The authors have done a good job in addressing my concerns and I am happy to accept the paper as it stands.
---

REVIEWER	Andrew Cook University of Southampton
REVIEW RETURNED	21-May-2017

GENERAL COMMENTS	This resubmission has generally addressed the concerns I raised previously. Two comments:
--

	(1) In the section on selective reporting, reference made to the protocol for the PASPORT study (36). The reference has however been updated to point to the in press trial report. I'd prefer at this point the the published protocol be referenced. (2) In the conclusions of the main discussion, I think the authors miss the distinction between efficacy and effectiveness. Efficacy is whether something works at all. Effectiveness is whether it works in everyday practice. However that's probably not the way I'd prefer to frame the research gaps. I'd suggest the important distinction is whether future trials should be explanatory or pragmatic, and whether outcome measures can be purely clinical or should be patient reported. This could probably best be set out by suggesting one more more questions in PICO format such as (and I'm making this up and not remotely suggesting this is a sensible question) P: patients having aortic valve replacement I: use of NIRS, C: usual care, O: preoperative stroke. -> What is the clinical and cost effectiveness of NIRS in the prevention of preoperative stroke in patients undergoing open aortic valve replacement? (The specified outcomes would push towards explanatory or pragmatic trials - e.g. serum creatine -> explanatory, need for dialysis -> pragmatic.)
--	--

VERSION 2 – AUTHOR RESPONSE

Reviewer: 2

This resubmission has generally addressed the concerns I raised previously.

Two comments:

(1) In the section on selective reporting, reference made to the protocol for the PASPORT study (36). The reference has however been updated to point to the in press trial report. I'd prefer at this point the published protocol be referenced.

The PASPORT trial paper is due to be published within the next 2 weeks and will be available as and when the manuscript goes to the publisher; the DOI for the manuscript is now provided for reference 36 to permit an accurate citation.

(2) In the conclusions of the main discussion, I think the authors miss the distinction between efficacy and effectiveness. Efficacy is whether something works at all. Effectiveness is whether it works in everyday practice. However that's probably not the way I'd prefer to frame the research gaps. I'd suggest the important distinction is whether future trials should be explanatory or pragmatic, and whether outcome measures can be purely clinical or should be patient reported. This could probably best be set out by suggesting one more more questions in PICO format such as (and I'm making this up and not remotely suggesting this is a sensible question) P: patients having aortic valve replacement I: use of NIRS, C: usual care, O: preoperative stroke. -> What is the clinical and cost effectiveness of NIRS in the prevention of preoperative stroke in patients undergoing open aortic valve replacement?

(The specified outcomes would push towards explanatory or pragmatic trials - e.g. serum creatine -> explanatory, need for dialysis -> pragmatic.)

Thank you for this comment the final sentence of the conclusion has been reworded to reflect this.

Reviewer: 1

Reviewer Name: DR SEEMA AGARWAL

Institution and Country: LIVERPOOL HEART AND CHEST HOSPITAL Please state any competing interests or state 'None declared': NONE DECLARED

Please leave your comments for the authors below The authors have done a good job in addressing my concerns and I am happy to accept the paper as it stands.

Thank you for this comment

Therefore, on behalf of my co-author, I am submitting the revised manuscript for consideration and possible publication. All authors have read and approved the submitted manuscript, the manuscript has not been submitted elsewhere nor published elsewhere in whole or in part.

VERSION 3 - REVIEW

REVIEWER	Andrew Cook Wessex Institute, , University of Southampton, UK
REVIEW RETURNED	25-Jun-2017

GENERAL COMMENTS	This has now address all my previous queries.
---